RelocaTE2: a high resolution transposable element insertion site mapping tool for population resequencing

Chen Jinfeng 1 2 3
Wrightsman Travis R. 3
Wessler Susan R. 2 3
Stajich Jason E. jason.stajich@ucr.edu 1 2
1 Department of Plant Pathology & Microbiology, University of California , Riverside , CA , United States
2 Institute for Integrative Genome Biology, University of California , Riverside , CA , United States
3 Department of Botany and Plant Sciences, University of California , Riverside , CA , United States
Mikheyev Alexander
Electronic publication date: 2017 Jan 26
Publication date: 2017
Volume: 5
Electronic Location ID: e2942
Received 2016 Sep 20; Accepted 2016 Dec 26
Copyright: ©2017 Chen et al.
Copyright year: 2017
Copyright holder: Chen et al.
License: This is an open access article distributed under the terms of the Creative Commons Attribution License, which permits unrestricted use, distribution, reproduction and adaptation in any medium and for any purpose provided that it is properly attributed. For attribution, the original author(s), title, publication source (PeerJ) and either DOI or URL of the article must be cited.
License URL: https://creativecommons.org/licenses/by/4.0/

Keywords: Annotation, Diversity, Parallel processing, Transposons, Population genomics, Short read, Bioinformatics, Rice, Resequencing

Funding: US National Science Foundation IOS-1027542 W. M. Keck Foundation Howard Hughes Medical Institute 52008110 US Department of Agriculture 2013-38422-20955 This research is supported by US National Science Foundation grant IOS-1027542 and a grant from the W. M. Keck Foundation to SR Wessler and JE Stajich and funds to SR Wessler from the Howard Hughes Medical Institute 52008110 and US Department of Agriculture Grant 2013-38422-20955. The funders had no role in study design, data collection and analysis, decision to publish, or preparation of the manuscript.

==============================
Background

Transposable element (TE) polymorphisms are important components of population genetic variation. The functional impacts of TEs in gene regulation and generating genetic diversity have been observed in multiple species, but the frequency and magnitude of TE variation is under appreciated. Inexpensive and deep sequencing technology has made it affordable to apply population genetic methods to whole genomes with methods that identify single nucleotide and insertion/deletion polymorphisms. However, identifying TE polymorphisms, particularly transposition events or non-reference insertion sites can be challenging due to the repetitive nature of these sequences, which hamper both the sensitivity and specificity of analysis tools.

Methods

We have developed the tool RelocaTE2 for identification of TE insertion sites at high sensitivity and specificity. RelocaTE2 searches for known TE sequences in whole genome sequencing reads from second generation sequencing platforms such as Illumina. These sequence reads are used as seeds to pinpoint chromosome locations where TEs have transposed. RelocaTE2 detects target site duplication (TSD) of TE insertions allowing it to report TE polymorphism loci with single base pair precision.

Results and Discussion

The performance of RelocaTE2 is evaluated using both simulated and real sequence data. RelocaTE2 demonstrate high level of sensitivity and specificity, particularly when the sequence coverage is not shallow. In comparison to other tools tested, RelocaTE2 achieves the best balance between sensitivity and specificity. In particular, RelocaTE2 performs best in prediction of TSDs for TE insertions. Even in highly repetitive regions, such as those tested on rice chromosome 4, RelocaTE2 is able to report up to 95% of simulated TE insertions with less than 0.1% false positive rate using 10-fold genome coverage resequencing data. RelocaTE2 provides a robust solution to identify TE insertion sites and can be incorporated into analysis workflows in support of describing the complete genotype from light coverage genome sequencing.

Introduction

Transposable elements (TE), mobile DNA of the genome, are drivers of genomic innovation (Bennetzen & Wang, 2014; Cordaux & Batzer, 2009). They can act as mutagens to disrupt gene functions or induce novel gene functions by providing enhancers or promoters that alter host gene expression (Feschotte, 2008; Lisch, 2013). In plants, TEs have been shown to contribute to several key trait innovations in crop domestication (Lisch, 2013). Systematic analysis of TE insertions and gene expression also suggests widespread roles of TEs in altering gene regulation (Kunarso et al., 2010; Lynch et al., 2011; Sundaram et al., 2014). It was found that 600–2,000 genetic variants between individuals in the human population and 200–300 variants between Arabidopsis accessions could be attributed to TE polymorphisms (Quadrana et al., 2016; Stewart et al., 2011). Although the magnitude of these polymorphisms is small compared to SNPs or other insertion/deletions, some TE polymorphisms are proximal to protein coding genes and can have large impacts on gene function or gene regulation (Cowley & Oakey, 2013; Quadrana et al., 2016; Stewart et al., 2011).

Two categories of bioinformatics tools have been developed to identify TE polymorphisms from population resequencing data. One type employs a strategy similar to that used to discover structural variations. These tools identify discordant pairs of sequence reads based on the chromosomal position of read alignments to indicate genomic inversions, insertions, deletions or other complex rearrangements (Campbell et al., 2008; Korbel et al., 2007). Software for TE mapping scrutinize genomic loci with discordant read pairs to see if known TE sequences are can be implicated near the rearrangement site. These tools, such as Retroseq (Keane, Wong & Adams, 2013) and TEMP (Zhuang et al., 2014), are generally highly sensitive and can locate TE insertions and absences to a 10–50 bp resolution. A second category of tools operates by first identifying by similarity any sequence reads containing partial or complete known TE sequences. Any sequence containing a TE is a “junction-read” which contains partial TE and partial unique host genomic sequence. These tools eliminate the TE sequence from these junction-reads and search the remaining 5′ or 3′ flanking sequence against the host organism genome sequence to identify the element’s location. These junctions-based tools, including RelocaTE (Robb et al., 2013) and ITIS (Jiang et al., 2015), are able to detect the exact location and TSD characteristic of TE insertion sites. This second category of tools is ideal for identifying new insertions from population resequencing data because it can accurately detect an insertion location along with the TSD. However, most of these tools are designed to search a single transposable element representative sequence at a time, which sacrifices speed for increased sensitivity and specificity. The extended runtime limits the feasibility of applying these tools when searching thousands of TEs in hundreds or thousands of individuals.

RelocaTE2 is an improved version of RelocaTE where we have implemented a junction-based approach that can search multiple template TEs in the same pass through short read sequencing data, streamlining the computational approach. Using simulated datasets, we show that RelocaTE2 is highly sensitive even in low coverage resequencing data or on chromosomes with a high repetitive sequence content with a specificity of greater than 99%. In comparing performance of related tools, RelocaTE2 was the most sensitive and specific tool in our tests profiling human and rice population genomics data. The tool is presented as a useful resource for analyzing population dynamics of TEs in genomic resequencing data.

Figure 1 Workflow for identification of transposable element insertions in population resequencing data using Illumina paired-end reads.

Materials & Methods

RelocaTE2 workflow

RelocaTE2 is based on the previous algorithm implemented in RelocaTE (Robb et al., 2013), which uses junction reads to find insertion sites of TEs. In RelocaTE2, we re-implement the search strategy to enable identification of multiple TEs in a single search, greatly increasing the speed and enabling searches for hundreds or thousands of candidate TE families in a genome (Fig. 1). We also implement new features in the algorithm to automatically identify TSDs and remove false junction reads (Fig. 1).

Briefly, the workflow initiates by matching a library of known repeat elements against short sequence reads generated by next generation sequencing, typically Illumina paired-end reads, using BLAT with the sensitive setting “-minScore = 10 -tileSize = 7” (Kent, 2002; Robb et al., 2013). Every read with similarity to repeat elements is denoted as an informative read and will contain a partial or complete copy of a TE. Informative reads that contain partial matches at the boundaries of the repeat elements are trimmed to remove the TE region so that only the regions flanking the element remain in either one or both of the paired-end reads (denoted as junction reads). Untrimmed versions of each junction read and its pair (denoted as full reads) are used as controls to filter false positive junction reads.

Sequence reads comprised entirely of repeat elements are ignored, but their read pair is kept (denoted as supporting reads). These junction, full, and supporting reads, are all aligned to the reference genome using BWA (v0.6.2) with the default setting “-l 32 -k 2” (Li & Durbin, 2009). Mapped reads are sorted by chromosome order and windows of 2,000 bp are evaluated to define insertion clusters. In each insertion cluster, additional subclusters are further refined based on the mapping position of junction reads to address the possible scenario of multiple insertion sites within a window. TSD position and sequence are identified if the subcluster is supported by junction reads from both upstream and downstream of the TE insertion site.

Next, a series of cleaning steps are used to filter low quality candidate insertion sites: (i) remove insertion sites that are only supported by low quality junction reads (map quality < 29); (ii) remove insertion site only supported by less than 3 junction reads total on the left or right flank when there are additional insertion sites which pass these filters in the same window. (iii) remove insertion sites only supported by junction reads and located within 10 bp range of an annotated TE in the reference genome. RelocaTE2 output reports the number of junction reads and supporting reads from both upstream and downstream of candidate TE insertion sites. Only confident insertions, defined as having at least one supporting junction read flanking the upstream or downstream of insertion sites and at least one junction read or one supporting read supporting the other end of TE insertion, are provided in the default output: “ALL.all_nonref_insert.gff”. Additional information about all candidate sites are provided in alternative output file: “ALL.all_nonref_insert.raw.gff”.

Simulated data for evaluation of TE insertion tools

Simulated datasets were created by randomly inserting TEs into sequence records of chromosomes 3 (OsChr3) and 4 (OsChr4) of rice (Oryza sativa L. ssp. japonica). OsChr3 is primarily made up of euchromatic regions, whereas OsChr4 has the sequence complexity consistent with heterochromatic regions and is a typical feature of many plant genomes (Zhao et al., 2002). Fourteen TEs families found in rice genomes comprised of 7 DNA Transposons: mPing, nDart, Gaijin, spmlike, Truncator, mGing, nDarz and 7 RNA Retrotransposons: Bajie, Dasheng, Retro1, RIRE2, RIRE3, Copia2, karma, were used. The insertion simulations were performed by choosing 200 random insertion sites on each chromosome in three independent replicates. Each simulated insertion site was generated by selecting one random chromosome position, one random TE, and inserting the element along with the expected TSDs. After generating 200 random insertions, a new genome sequence was generated along with a GFF3 file containing the recorded insertion locations to support the performance evaluation of the dataset. Paired-end reads of all simulated chromosomes were simulated by pIRS (pirs simulate –l 100 –x coverage –m 500 –v 100) (Hu et al., 2012). For each dataset, simulate sequence reads at sequence depths of 1, 2, 3, 4, 5, 6, 7, 8, 9, 10, 15, 20, 40-fold coverage were generated.

Real sequence data for evaluation of TE insertion tools

Three sets of data, an individual human genome: HuRef, an individual rice genome: IR64, and population resequencing data of 50 rice and wild rice genomes (Levy et al., 2007; Schatz et al., 2014; Xu et al., 2012), were used to evaluate the performance of RelocaTE2 and TEMP. The high quality genome assemblies of HuRef and IR64 were used to evaluate the accuracy of TE genotyping tools by comparing each to a reference genome assembly for the species assembled. The HuRef (also known as Venter genome) has been extensively studied for TE insertions (Xing et al., 2009). Previous work identified 574 Alu elements that have been experimentally verified and can be treated as a gold standard data set for evaluation (Hormozdiari et al., 2010; Xing et al., 2009). Paired-end sequence reads of 10-fold depth were simulated from HuRef as test dataset by pIRS (pirs simulate -l 100 -x coverage -m 500 -v 100) (Hu et al., 2012). RelocaTE2 and TEMP were tested and their results compared to the Human Genome Reference Consortium genome (GRCh36 or hg18). A second dataset, the finished reference genome assembly of rice strain IR64, was explored utilizing available Illumina sequencing reads (Schatz et al., 2014). RelocaTE2 and TEMP were tested on 20-fold genome coverage of 100 bp paired-end Illumina short reads (SRA accession: SRR546439) aligned to the rice reference genome (MSU7). A third dataset was composed of resequencing data from 50 strains of rice and wild rice population with an average sequencing depth of 17-fold (Xu et al., 2012). RelocaTE2 and TEMP were tested on the sequencing libraries from each of these 50 strains to assess their consistency across datasets with varying sequence depth and genetic diversity. RelocaTE and ITIS were not included in the biological data testing because of the prohibitively long run times on these large datasets and their poor performance on simulated datasets.

Detection of TE insertions using RelocaTE2, RelocaTE, TEMP and ITIS

RelocaTE2, RelocaTE, TEMP and ITIS were run with default parameter settings on simulated data. The results were filtered to evaluate the best performing parameters for each tool. RelocaTE2 was tested with parameters “–len_cut_match 10 –len_cut_trim 10 –mismatch 2 –aligner blat”, which uses BLAT as the search engine (–aligner blat), allows for 2 mismatches (–mismatch 2) in matched sequence between reads and repeat elements (–len_cut_match 10) and only keeps sequence fragments that have at least 10 bp after trimming repeat elements from reads (–len_cut_trim 10). RelocaTE was tested using parameters “–len_cutoff 10 –mismatch 0”, which uses BLAT as search engine by default and allowed 0 bp mismatch (–mismatch 0) for matched sequence between reads and repeat elements (–len_cutoff 10). It should be noted that the mismatch setting in RelocaTE is the ratio of base pairs in the alignment between reads and repeat elements that can be mismatched, not an integer number of allowed mismatches, as used in RelocaTE2. Singleton calls from RelocaTE’s results, which are sites supported by only one read, were removed. TEMP was tested with parameters “-m 3”, which allow for three mismatches between reads and repeat elements. Singleton calls from TEMP’s results were removed when testing on simulated data to achieve a balance between sensitivity and specificity. ITIS was tested with default parameters, which filtered TE calls with at least one read supporting from both ends of TE insertions. For analysis of the HuRef genome, the IR64 genome and the 50 rice and wild rice strains, RelocaTE2 and TEMP were run with default parameter settings as described above. The TEMP results were filtered to keep only TE calls with supporting and/or junction reads from both ends of TE insertions to achieve a comparable balance between sensitivity and specificity.

Results and Discussions

Performance of RelocaTE2, RelocaTE, TEMP and ITIS on simulated data

RelocaTE2 was first compared to RelocaTE, TEMP and ITIS using the simulated datasets. Each dataset of simulated rice chromosomes, OsChr3 and OsChr4, was virtually sheared to simulate paired-end short reads at a coverage ranging from 1 to 40-fold. At high sequencing coverage (≥10-fold), RelocaTE2, TEMP and ITIS were able to identify >99% of simulated insertions on OsChr3, whereas the performance of RelocaTE was much lower (85%) (Fig. 2A). At lower sequencing coverage, e.g., 3-fold, only RelocaTE2 and TEMP were able to achieve ≥95% sensitivity on OsChr3 (Fig. 2A). Furthermore, TEMP was able to identify 83% and 93% of simulated insertions on OsChr3 at very low sequence coverage of 1-fold and 2-fold, respectively (Fig. 2A). RelocaTE2 had a sensitivity of 53% and 83% on OsChr3 for the 1-fold and 2-fold coverage due to the removal of TE insertions supported by only one read (singleton) or supported by reads from only one end of TE insertions (insufficient insertions), which can result in many false positives (Fig.  2A).

Figure 2 Performance of RelocaTE2, RelocaTE, TEMP and ITIS on simulated rice data.

Comparison of tool performance on rice chromosome 3 (OsChr3) for Sensitivity (A), Specificity (B), Recall rate of Target Site Duplication (TSD) (C), and comparison of performance on rice chromosome 4 (OsChr4) for Sensitivity (D), Specificity (E), Recall rate of TSD (F). Three replicate simulations of 200 random transposable element (TE) insertions were generated for OsChr3 and OsChr4. A series of datasets were constructed by sampling at varying sequence depths (from 1 to 40) from each simulated TE datasets. Sensitivity (SN), Specificity (SP) and TSD recall of each tool was estimated on each simulated dataset across multiple sequence depths. The error bars show the standard deviation among the three replicates which had different composition of 200 random TE insertions. SN was defined as the percentage of TE insertions from 200 simulated TE insertions were recalled within 100 base pairs of simulated TE insertion sites. SP was defined as the percentage of TE insertions from all calls were within 100 base pairs of 200 simulated TE insertions. Recall rate of TSD was defined as the percentage of true positive calls that correctly matched the simulated TSD of TE insertions.

RelocaTE2, RelocaTE and TEMP showed >99% specificity on OsChr3 at multiple levels of sequence coverage (Fig. 2B). In contrast, the specificity of ITIS was much lower (<90%), even when run on the high sequence coverage dataset on OsChr3 (Fig. 2B). In comparing recall rates of TSDs, RelocaTE2 and ITIS had similar performance and achieved the highest recall rate of 98% and 91% respectively, on OsChr3 at ∼10-fold coverage (Fig.  2C). The recall rate of TSDs for both TEMP and RelocaTE depended on sequence depth and achieved only 37% and 60%, respectively, at 10-fold coverage (Fig.  2C). All the tools performed worse on OsChr4 as compared to OsChr3 (Figs. 2D–2F). RelocaTE2 demonstrated a lower average sensitivity (92%) on OsChr4 when compared OsChr3 (96%) (Figs. 2A–2D). Similarly, TEMP had a slightly lower sensitivity (95%) on OsChr4 than on OsChr3 (97%) (Figs. 2A–2D). However, RelocaTE2 and RelocaTE demonstrated high level of the specificity (>99%) while TEMP performed at a slightly lower specificity (98%) on OsChr4 compared to >99% on OsChr3 (Figs. 2B and 2E). In comparing TSD accuracy on OsChr4, on average 81% of RelocaTE2 calls correctly identified the TSD, whereas only 31% of TEMP calls were correct (Figs. 2C and 2F).

Evaluation of RelocaTE2 and TEMP on biological datasets

We evaluated TE identifying tools in the HuRef genome and benchmark the sensitivity and specificity of these tools using 574 experimental verified Alu insertions in HuRef genome and genomic comparison between HuRef genome and GRCh36. RelocaTE2 and TEMP reported similar results and identified 83% (479/574) and 76% (438/574) of standard insertion sites (Fig. 3A). Comparing the HuRef genome with GRCh36 suggested that 89% and 95% of insertions identified by RelocaTE2 and TEMP, respectively, were real insertions (Fig. 3A). In addition, RelocaTE2 predicted TE insertion sites with higher precision (9 ± 6 bp) compared to TEMP (366 ± 170 bp).

Figure 3 Performance of RelocaTE2 and TEMP on biological dataset in HuRef genome, IR64 genome, and 50 rice and wild rice strains.

(A) Venn diagram of the overlap in non-reference TE insertions identified in the HuRef genome and the rice IR64 genome using RelocaTE2 and TEMP. Sensitivity (SN) and Specificity (SP) were assessed by comparing the assembled HuRef genome to the GRCh36 reference genome and the assembled IR64 genome to the MSU7 reference genome. SN was defined as the percentage of validated calls out of all validated calls by either RelocaTE2 or TEMP. SP was defined as the percentage of validated calls out of all calls by each tool. (B) Comparison of the number of non-reference TE insertions of 14 TE families in 50 rice and wild rice strains identified by RelocaTE2 and TEMP. Strains are color-coded based on subpopulation classification.

RelocaTE2 and TEMP were used to analyze data from the rice strain IR64 and the results were evaluated by comparing the genome assembly of IR64 with MSU7. RelocaTE2 identified 648 insertion sites while the genome comparison revealed that 93% of insertions were true positives (Fig.  3A). TEMP identified 362 insertions, of which 50% (183/362) overlapped with RelocaTE2 (Fig.  3A). The specificity of TEMP was estimated to be 86%, slightly lower than RelocaTE2 (93%) (Fig.  3A). However, TEMP was found to be less sensitive than RelocaTE2 in the rice genome, only calling 362 sites as compared to 648 by RelocaTE2 (38% vs. 90%, Fig. 3A). Moreover, RelocaTE2 predicted TE insertion junctions of 3 ± 1 bp, which was much smaller than TEMP (393 ± 199 bp).

RelocaTE2 and TEMP were used to identify TE polymorphisms in 50 resequenced rice and wild rice strains, which contain substantial sequence diversity and population structure (Xu et al., 2012). The results from these two tools were well correlated (R2 = 0.96, P value = 2.2e–16) and predicted more TE insertions in the diverged population of wild rice,O. nivara and O. rufipogon, and even in the indica population than japonica rice which close to the reference genome (Fig. 3B). On average 72% of the sites predicted in these 50 rice and wild rice strain by RelocaTE2 and TEMP overlapped. Many insertion sites from TEMP were predicted with only supporting read flanking one end of an insertion, which produced large variations in predicted junctions of TE insertion sites (118 ± 151 bp). In contrast, RelocaTE2 reported most of TE insertions supported by junction reads or supporting reads on both ends, which resulted in accurate insertion junction predictions (3 ± 2 bp).

Runtime performance

We implemented the searching process for TE insertion to run on multiple processors in Python. The process is relatively memory efficient. When searching TEs in the rice genome for example, one process generally uses less than 1 Gb memory. The running time of RelocaTE2 depends on number of processors used. Searching 3,000 templates of transposable elements with 20-fold genome coverage sequencing data of the rice genome takes 3–4 h for RelocaTE2 using 32 CPUs including the alignment steps. TEMP identifies transposable element insertions from a BAM file. It takes ∼1 h for TEMP for the same project using single process. RelocaTE (version 1) and ITIS take at least days for the same rice datasets and can be prohibitively difficult to run on large datasets with multiple templates due to the serial searching approach of their implementation.

Conclusions

We present RelocaTE2 as a new tool for mapping TE insertions to base-pair resolution from resequencing data. RelocaTE2 identifies multiple TE families in a single search with high sensitivity and specificity. The evaluation of these tools on simulated and biological datasets support the use of RelocaTE2 for analysis of genomes of plants and animals and indicate it can generate very high quality genotyping of TE insertions from resequencing datasets of modest sequencing depths. TEMP is fast and achieves well balance between sensitivity and specificity in identifying TE polymorphisms, particularly when the sequence depth is shallow. Although the precision of TSD prediction of TEMP is not very sensitive compared to RelocaTE2 and other tools. We recommend using TEMP in scenarios when accuracy of TSD is not very critical. The high resolution mapping of TE insertions sites will enable detailed analysis of the interaction of TEs and genes and as structural variations that vary in populations.

Supplemental Information

Supplemental Information 1 RelocaTE2 Source code

Release source code for version of RelocaTE2 described in this manuscript.

Click here for additional data file.

Additional Information and Declarations

Competing Interests

Author Contributions

Data Availability

The authors declare there are no competing interests.

Jinfeng Chen conceived and designed the experiments, performed the experiments, analyzed the data, wrote the paper, prepared figures and/or tables, reviewed drafts of the paper.

Travis R. Wrightsman performed the experiments, contributed reagents/materials/analysis tools, reviewed drafts of the paper.

Susan R. Wessler and Jason E. Stajich conceived and designed the experiments, wrote the paper, reviewed drafts of the paper.

The following information was supplied regarding data availability:

RelocaTE2 code repository (https://github.com/stajichlab/RelocaTE2) and archived at Zenodo: http://dx.doi.org/10.5281/zenodo.208123.

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
