# Peer review of "RelocaTE2: a high resolution transposable element insertion site mapping tool for population resequencing"

_PeerJ, doi:10.7717/peerj.2942_

## Round 0.1 · original submission · Minor Revisions

The reviewers were in agreement that the study is well-designed and the manuscript is well-written. Each of them has a number of specific minor changes that they would like fixed prior final acceptance.

Reviewer 1 ·

Basic reporting

Chen et al. describes a new bioinformatics tool for the identification of genetic polymorphisms in Transposable Elements (TE). RelocaTE2 is based on the tools that first identify TE by similarity and is ideal for identifying new insertions from population sequencing. I appreciate and give a positive response to this work. However, my first recommendation is the change of title. When I read the title for the first time, I imagined a software to identify all types of TE sequence polymorphisms (e.g., SNPs), but the tool only identifies a one type of polymorphism (insertion).

Experimental design

The positive point of tool is the ability to identify TE polymorphisms with high sensitivity and speed in low deep-sequencing coverage data. This is a positive point of tool. However, for a user without computer training (e.g., agronomists) it is almost impossible to install the software. I recommend using a system that streamlines this process, such as miniconda (http://conda.pydata.org/miniconda.html).

Minor questions:
82 → TDS? What is TDS?
93 → What is the size of junctions? Are they long-reads? Why not use the BWA-MEM?
144 → How many sequences?
387 → Low quality figures. It is impossible to interpret any results. List the letters (A,B,C,D,E,F) of each graph with the legend.

Validity of the findings

No comments.

Additional comments

No comments.

·

Basic reporting

The introduction is comprehensive and sets the paper into context well. Overall the paper is clearly written, I have only a couple of specific points to make.
From my experience I believe that TEMP combines the discordant pair and junction based approaches and so on occasion is capable of accurately determining the target site duplication of a transposable element insertion. Though this feature of TEMP, as shown in the results presented here, is not always accurate, it is important to acknowledge it.
On line 69 the phrase “junction-based approach” is used for the first time. It may be useful to those less familiar with the field to use this term around lines 59 to 61 where this category of method is described.
On lines 133-135 the authors say "High quality reference genome assemblies of HuRef and IR64... comparing the assembled sequences to the reference genome.". Removing "reference" from the start of the sentence could make what was done here a bit clearer.

Experimental design

The method is well explained and parameters used are clearly discussed. The combined simulation and real data approach to evaluating the performance of the method is logical and comprehensive and so allows the evaluation of the methods tested. I was pleased to find that the software written here was available on GitHub with appropriate instructions for installation and use. The software not only installed without problem, but also included copies of the correct versions of all dependencies. Problems with dependencies can take a very long time to fix and so this is a welcome aid to potential users. The inclusion of test data was also welcome and allowed me to quickly confirm the installation had worked and the software was functional.

Validity of the findings

The results presented clearly demonstrate the validity of the RelocaTE2 method. To satisfy my curiosity I ran RelocaTE2 on some saccharomyces cerevisiae data that I had previously used six other TE detection methods on. I can confirm that in my tests RelocaTE2 does indeed outperform TEMP and the previous version of RelocaTE for both sensitivity and specificity, as shown in the paper. In addition, RelocaTE2 also outperforms the four other (older) methods I had used.

Reviewer 3 ·

Basic reporting

The article fulfill the standards.

Experimental design

The article fulfill the standards.

Validity of the findings

The article fulfill the standards.

Additional comments

The manuscript describes and tests a new version of a tool designed to map transposable element polymorphisms using resequencing data: RelocaTE2. The authors evaluate the performance of their software by comparing the specificity and sensitivity to the previous version of the same software and two additional tools. I have three comments:
1. In the introduction section. The authors mention that both RelocaTE and T-lex2, among other tools, are ideal for identifying new insertions from population resequencing data. I am not sure whether I understand what the authors mean by "new insertions". T-lex2 requires the annotation of the TEs and evaluates whether that particular TE is present, absent or polymorphic in a given NGS dataset. As such, it does not identify new insertions, it checks the frequency of TEs already annotated in a reference genome.
2. Figure 2B. The specificity for the RelocaTE2 tool can not be seen in the graph. I am guessing it is because it totally overlaps with one of the other datasets. Since, this is the tool described in this paper, maybe plot the result for RelocaTE2 and add to the figure legend that the other dataset can not be seen in the graph because it overlaps with RelocaTE2.
3. In the abstract, the authors mention that the RelocaTE2 demonstrates a higher level of sensitivity and specificity when compared to other tools. That´s only true in some scenarios, and should be mentioned accordingly. However, it looks like RelocaTE2 outcompetes other softwares in the precision with which insertion sites are defined. I think this is very important since it will allow to get good frequency estimates when more than one dataset is available for the same species.

---

## Round 0.2 · accepted · Accept

The authors have responded to all of the minor criticisms raised by the reviewers.